# Measuring the fitted filtration efficiency of cloth masks, medical masks and respirators

**Amanda A. Tomkins[1], Gurleen Dulai[1], Ranmeet Dulai[2], Sarah Rassenberg[1], Darren Lawless[3], Scott Laengert[1,3], Rebecca S. Rudman[4], Shiblul Hasan[5], Charles-Francois de Lannoy[1,3], Ken G. Drouillard[6], Catherine M. Clase [3,7]***

1 Department of Chemical Engineering, McMaster University, Ontario, Canada, 2 Faculty of Health Sciences, Queen's University, Ontario, Canada, 3 Centre of Excellence in Protective Equipment and Materials, McMaster University, Ontario, Canada, 4 Windsor-Essex Sewing Force, Windsor, Ontario, Canada, 5 School of Computer Science, University of Windsor, Windsor, Ontario, Canada, 6 Great Lakes Institute for Environmental Research, University of Windsor, Windsor, Ontario, Canada, 7 Department of Medicine, Faculty of Health Sciences, McMaster University, Ontario, Canada,

* clase@mcmaster.ca

## Abstract

### Importance

Masks reduce transmission of SARS-CoV2 and other respiratory pathogens. Comparative studies of the fitted filtration efficiency of different types of masks are scarce.

### Objective

To describe the fitted filtration efficiency against small aerosols (0.02–1 µm) of medical and non-medical masks and respirators when worn, and how this is affected by user modifications (hacks) and by overmasking with a cloth mask.

### Design

We tested a 2-layer woven-cotton cloth mask of a consensus design, ASTM-certified level 1 and level 3 masks, a non-certified mask, KF94s, KN95s, an N95 and a CaN99.

### Setting

Closed rooms with ambient particles supplemented by salt particles.

### Participants

12 total participants; 21–55 years, 68% female, 77% white, NIOSH 1–10.

### Main Outcome and Measure

Using standard methods and a PortaCount 8038, we counted 0.02–1 µm particles inside and outside masks and respirators, expressing results as the percentage filtered by each mask. We also studied level 1 and level 3 masks with earguards, scrub caps, the knot-and-tuck method, and the effects of braces or overmasking with a cloth mask.

**Data availability statement:** Data are available in a public repository at https://osf.io/58g2j/

**Funding:** The author(s) received no specific funding for this work.

**Competing interests:** Amanda Tomkins is a member of Dr Qiyin Fang's research group which worked on the silicone mask brace. She is also a member of the cloth mask knowledge exchange, a stakeholder group that includes cloth mask manufacturers and fabric distributors. Catherine Clase has received consultation, advisory-board membership, honoraria, or research funding from the Ontario Ministry of Health, Sanofi, Pfizer, Leo Pharma, Astellas, Janssen, Amgen, Boehringer-Ingelheim, Baxter and, through LiV Academy, AstraZeneca. In 2018 she co-chaired a KDIGO potassium controversies conference sponsored at arm's length by Fresenius Medical Care, AstraZeneca, Vifor Fresenius Medical Care, Relypsa, Bayer HealthCare and Boehringer Ingelheim. She co-chairs the cloth mask knowledge exchange, a stakeholder group that includes cloth mask manufacturers and fabric distributors. She is editor-in-chief of MaskEvidence.org. Ken G Drouillard is a member of the WE-SPARK Health Institute, University of Windsor and receives funding from the Natural Sciences and Engineering Research Council of Canada, Environment and Climate Change Canada and Ontario Ministry of Conservation, Environment and Parks. In 2020-2022 he acted as science consultant to the Windsor-Essex Sewing Force, a community group engaged in the design, sewing and donation of cloth masks to healthcare providers and vulnerable populations of Southern Ontario. He is a member of the cloth mask knowledge exchange. Charles-Francois de Lannoy has received funding from various branches of The Natural Sciences and Engineering Research Council of Canada (NSERC), Ontario Centre of Innovation (OCI), formerly Ontario Centres of Excellence (OCE), Ontario Water Consortium (OWC) formerly Southern Ontario Water Consortium (SOWC), Canada First Research Excellence Fund (CFREF), Ontario Together Fund, and Federal Economic Development Agency for Southern Ontario (FedDev). He is a member of cloth mask knowledge exchange, a stakeholder group that includes cloth mask manufacturers and fabric distributors. Darren Lawless co-chairs the cloth mask knowledge exchange, and all authors are members. Other authors have no additional disclosures. This does not alter our adherence to PLOS ONE policies on sharing data and materials.

## Results

Filtration efficiency for the cloth mask was 47–55%, for level 1 masks 52–60%, for level 3 masks 60–77%. A non-certified KN95 look-alike, two KF94s, and three KN95s filtered 57–77%, and the N95 and CaN99 97–98% without fit testing. External braces and over-masking with a well-fitting cloth mask increased filtration, but earguards, scrub caps, and the knot-and-tuck method did not.

## Limitations

Limited number of masks of each type sampled; no adjustment for multiple comparisons.

## Conclusions and Relevance

Well-fitting 2-layer cotton masks filter in the same range as level 1 masks when worn: around 50%. Level 3 masks and KN95s/KF94s filter around 70%. Over a level 1 mask, external braces or overmasking with a cloth-mask-on-ties produced filtration around 90%. Only N95s and CaN99s, both of which have overhead elastic, performed close to the occupational health and safety standards for fit tested PPE (>99%), filtering at 97–99% when worn, without formal fit testing. These findings inform public health messaging about relative protection from aerosols afforded by different mask types and explain the effectiveness of cloth masks observed in numerous epidemiologic studies conducted in the first year of the pandemic. A plain language summary of these findings is available at https://maskevidence.org/masks-compared.

## Introduction

The global shortage of personal protective equipment (PPE) during the COVID-19 pandemic motivated inquiry into the effectiveness of public use of cloth masks [1–19]. In July 2024, current WHO guidance recommends, "Make wearing a mask a normal part of being around other people" [20]. With increased manufacturing of masks and PPE, a variety of masks and respirators are available, including home-sewn and commercial cloth masks, certified medical masks and non-certified disposable masks, KN95/KF94s and N95/CaN99s.

Aerosol transmission of COVID-19 is now widely accepted [17,21–26]. Systematic evaluation of the filtration of aerosols, from the perspective of the wearer, across the whole range of masks and non-fit-tested respirators is highly relevant to public health advice. Descriptions of cloth masks as effective only against droplets [27] or only as source control reflects and perpetuates public misunderstanding. We extend knowledge in this area by using an adequately-powered sample and a variety of masks, and by fully characterizing participants, masks, materials and modifications.

## Methods

### Population and sampling strategy

Between March 2021 and April 2022, we recruited adults (≥16y), purposively sampling people of non-European ancestry, excluding people with respiratory diseases or allergic to latex. Investigators were included as participants. The Hamilton Integrated Research Ethics Board approved the study, safety procedures were developed, and participants gave informed consent.

## Procedures

**Facial measurement.** We measured participants' faces [28] according to the US National Institute of Occupational Health and Safety (NIOSH) Bivariate Panel (S1 Appendix) [29].

**Masks.** We studied level 1 masks (Polar Bear and O2) and level 3 masks (Halyard and Primed), certified to the standards of ASTM International [30–32] and donated by Hamilton Health Sciences, Hamilton, ON, Canada; these are worn on elastic earloops (S1 Fig). A two-layer pleated mask of woven cotton without a nosewire, the Essex mask, designed by a consensus panel for the Windsor-Essex Sewing Force (WESF) was studied both on ¼ inch elastic earloops and on overhead cloth ties (https://maskevidence.org/patternsinstruction) [33]. We purposively sampled non-certified masks, KN95s, KF94s (S2 Appendix). We purchased 3M N95 Aura 1870 + respirators from Steripro Canada PPE Store and Vitacore CaN99 respirators from Vitacore Industries Inc, Burnaby, BC. We did not formally fit-test the respirators using occupational health and safety protocols.

**Ambient particle count.** We used a TSI 8026 Particle Generator (TSI, Shoreview, MN), placed at least 2 m from the measuring device and participant. This aerosolises dissolved sodium chloride, producing a polydisperse aerosol that dries to solid sodium chloride particles with a diameter in the range of 0.02 μm to 0.60 μm, supplementing naturally-occurring particles in the room. We proceeded with testing if the total particle count was between 2000 and 20,000 particles/cm³. If the ambient particle count at the end of a test was not ± 30% of the count at the start, we retested.

**Filtration efficiency testing.** We fitted each mask with a sampling probe (a short aluminium tube with a flange) using a TSI model 8025-N95 Fit Test Probe Kit (TSI, Shoreview, MN). For cloth masks, if we could not penetrate the mask with the probe kit, we first used a scalpel to create a mask puncture of 1 mm. We placed the sampling port at the center of the mask; for masks with a central seam, we placed the port in the lateral flat surface. The probe attaches to a plastic tube that samples air from inside the mask. A second tube is suspended by a lanyard around the participant's neck so that its opening is at chest height. Air sampled through this second tube reflects ambient particle counts (justification and detailed methods given in S3 Appendix; S2 Fig).

Both tubes were connected to a PortaCount 8038 (TSI Inc, Shoreview, MN) [34]. We used CSA-Z94.4-2002 protocol in all-particles mode (i.e., with the N95 classifier off) to detect particles in the range 0.02–1 μm [34,35]. Participants donned masks and adjusted them to achieve the best subjective fit. Participants rated each mask or combination for subjective leak, and for glasses fog (using their own or safety glasses). After removing eyewear, participants conducted each of the following exercises, for 37 seconds each, according to protocol: normal breathing, deep breathing, head turning, head nodding, talking, bending, and a second period of normal breathing. Finally, participants rated each mask or combination for discomfort, using Likert scales with anchors that we developed for this study (S4 Appendix). Participants also reported specific issues with masks and provided subjective free-form comments.

Fit factor for each exercise is a dimensionless number, defined as the ratio of ambient to within-mask particle count, and calculated as the geometric mean of the fit factor across all exercises for one participant. We transformed individual fit factor value to fitted filtration efficiency, as a percentage [2], using:

$$\left(1 - \frac{1}{fit\ factor}\right) * 100\% \tag{1}$$

Fitted filtration efficiency describes the percentage of particles filtered by the mask. A fit factor of 4, reflecting four times the particles outside compared with inside, has a fitted filtration efficiency of 75%.

**Community and participant involvement.** The Cloth Mask Knowledge Exchange is a stakeholder group formed by the Centre of Excellence in Protective Equipment and Materials at McMaster in 2020 (S5 Appendix). This group gave input throughout the design and conduct of the experiments.

## Statistical methods

We used SYSTAT v13, Inpixon, Palo Alto, CA for statistical analyses. We analyzed the transformed variable filtration efficiency. We used parametric tests (ANOVA with post hoc Tukey's honestly significant difference) when data met normality assumptions by Lilliefors test; and non-parametric tests (Kruskal-Wallis with post-hoc Conover-Inman pairwise comparisons) when they did not, estimating mean and 95% confidence intervals or median and interquartile range, respectively, and regarding $P < 0.05$ as statistically significant.

**Studies.** Mask types. On 4 participants, we studied the filtration efficiency of cloth masks on earloops and on overhead ties, level 1 and level 3 certified medical masks (two of each, on earloops), a KN95, two KN94s, a non-certified KN-95-lookalike mask, an N95 and a CaN99 (S1 Fig).

Mask hacks. On 10 participants, we studied the filtration efficiency of ASTM level 1 (Polar Bear) and level 3 (Halyard) masks, worn as intended, and with minor modifications or hacks (S1 Fig) [5,36–38]. Three participants had short beards [39,40].

Overmasking. On 6 participants, in triplicate, we studied overmasking, using an Essex mask on earloops, a level 1 (Polar Bear) and a level 3 (Halyard) certified mask as the base mask, and overmasking with either an Essex mask on earloops or an Essex mask with overhead ties (S1 Fig).

**Power calculation.** In preliminary data, mean filtration efficiency was 50%, standard deviation 10%, for level 1 masks and cloth masks. For the study of mask types, the aim was to show the range of absolute filtration for each; using 4 participants gives 95% confidence intervals of 50–70% around a mean of 60% and 80–100% around a mean of 90%. For mask hacks and overmasking, we were interested in detecting differences of 10%; at alpha 0.05 and 80% power, we calculated that 6 people were needed to test each mask [41]. To improve power and generalizability, we recruited 10 participants to the study of mask hacks, and to improve power, we performed measurements in triplicate on 6 participants in the overmasking study.

## Results

The 12 participants were 21–55 years, 58% female, 25% non-European, NIOSH 1–10 (Table 1 and S1 Table). Data are shared in an online repository, OSFHOME at https://osf. io/58g2j/.

## Mask types

Tested on 4 participants, the 2-layer cotton pleated Essex mask resulted in a mean ± standard deviation (SD) fitted filtration efficiency of 47 ± 5% when worn on earloops and 55 ± 6% on overhead ties ($P > 0.05$; Fig 1). These were comparable with results for the certified level 1 masks, at 52 ± 6% and 56 ± 9%, and one of the level 3 masks at 60 ± 6% ($P > 0.05$), but a second level 3 mask filtered at 75 ± 10% ($P < 0.05$). The KF94 and KN95s, and the KN95-lookalike mask filtered between 57% and 77% (FFE range across trials and masks). The certified respirators (not fit-tested) filtered at 97–98% (FFE range across trials and masks). The lowest value for an individual for either respirator was 95%.

**Table 1. Summary demographic and anthropometric data on participants.**

| Study | Mask Types | Mask Hacks | Overmasking |
|---|---|---|---|
| Number of Participants | 4 | 10 | 6 |
| Number of Replicates | 1 | 1 | 3 |
| Age, y (mean ± SD) | 30 ± 17 | 35 ± 17 | 38 ± 18 |
| Women/Men | 4/0 | 5/5 | 5/1 |
| Facial hair (n, %) | 0 | 3 | 0 |
| Self-identified Ethnicity (n, %) | | | |
| European | 2 | 9 | 4 |
| West Central Asian and Middle Eastern | 2 | 1 | 2 |
| Height, m (mean ± SD) | 1.67 ± 0.08 | 1.76 ± 0.08 | 1.67 ± 0.08 |
| Weight, kg (mean ± SD) | 63 ± 12 | 77 ± 14 | 63 ± 16 |
| Bizygomatic distance, calipers, cm (mean ± SD) | 12.3 ± 0.9 | 13.7 ± 1.1 | 12.3 ± 1.1 |
| Menton-sellion distance, calipers, cm (mean ± SD) | 11.0 ± 0.8 | 11.5 ± 1.3 | 11.0 ± 1.0 |
| Bizygomatic distance, cord, cm (mean ± SD) | 15.7 ± 1.1 | 15.2 ± 2.3 | 15.7 ± 2.2 |
| Menton-sellion distance, cord, cm (mean ± SD) | 14.5 ± 2.5 | 15.5 ± 2.2 | 14.5 ± 2.0 |
| NIOSH Bivariate Panel Size (n, %) | | | |
| 1 (small) | 1 | 2 | 1 |
| 2-4 (medium) | 1 | 2 | 2 |
| 5-7, 9 (large) | *2 | 4 | *3 |
| 8, 10 (X-large) | 0 | 2 | 0 |

SD standard deviation. NIOSH related to small, medium, large, X-large sizing according to ASTM 3502:22a [63].

*Includes 2 participants with face width narrower than smallest face width on the NIOSH panel, classified in the nearest cell at NIOSH 6.

We studied participants with facial hair only in the mask hacks study. Facial hair was classified according to CDC NIOSH [39]

## Mask hacks

Tested on 10 participants, median filtration efficiency of the level 1 and level 3 mask was not improved by wearing on ear guards, scrub cap, or by knot-and-tuck ($P > 0.05$, Fig 2). Both masks were improved ($P < 0.05$) by braces designed to improve edge seal. For level 1 masks, best performance was realized using the silicone and Fix-The-Mask braces generating median (25–75 percentiles) FFEs of 89 (87–91)% and somewhat lower relative performance with the neoprene brace at 85 (79–87)%. For level 3 masks, the three tested braces yielded consistent results with median (25–75 percentile) FFEs 93 (92–94)% that were improved ($P < 0.05$) over baseline..

## Overmasking

Tested on 6 participants (3 replicates per individual), mean ± SD filtration efficiency for the Essex mask on earloops was 47 ± 6%, for the Essex on overhead ties 55 ± 5%, and the level 1 mask 52 ± 6% ($P > 0.05$); the level 3 mask filtered 70 ± 13% ($P < 0.05$; Fig 3). Overmasking the Essex-on-earloops with another on earloops did not improve filtration but overmasking with an Essex-on-ties improved filtration to 66 ± 6% ($P < 0.05$). For both the level 1 and level 3 masks, overmasking with an Essex mask improved filtration. The level 1 mask overmasked with Essex-on-ties (84 ± 10%) performed better than level-1–Essex-on-earloops (73 ± 12%), both exceeding the level 1 mask control (all $P < 0.05$). The level 3 mask over-masked with Essex-on-earloops (84 ± 13%) exceeded the control ($P < 0.05$) as did level-3–Essex-on-ties (92 ± 3%; p < 0.05), but the difference between the two was not statistically significant.

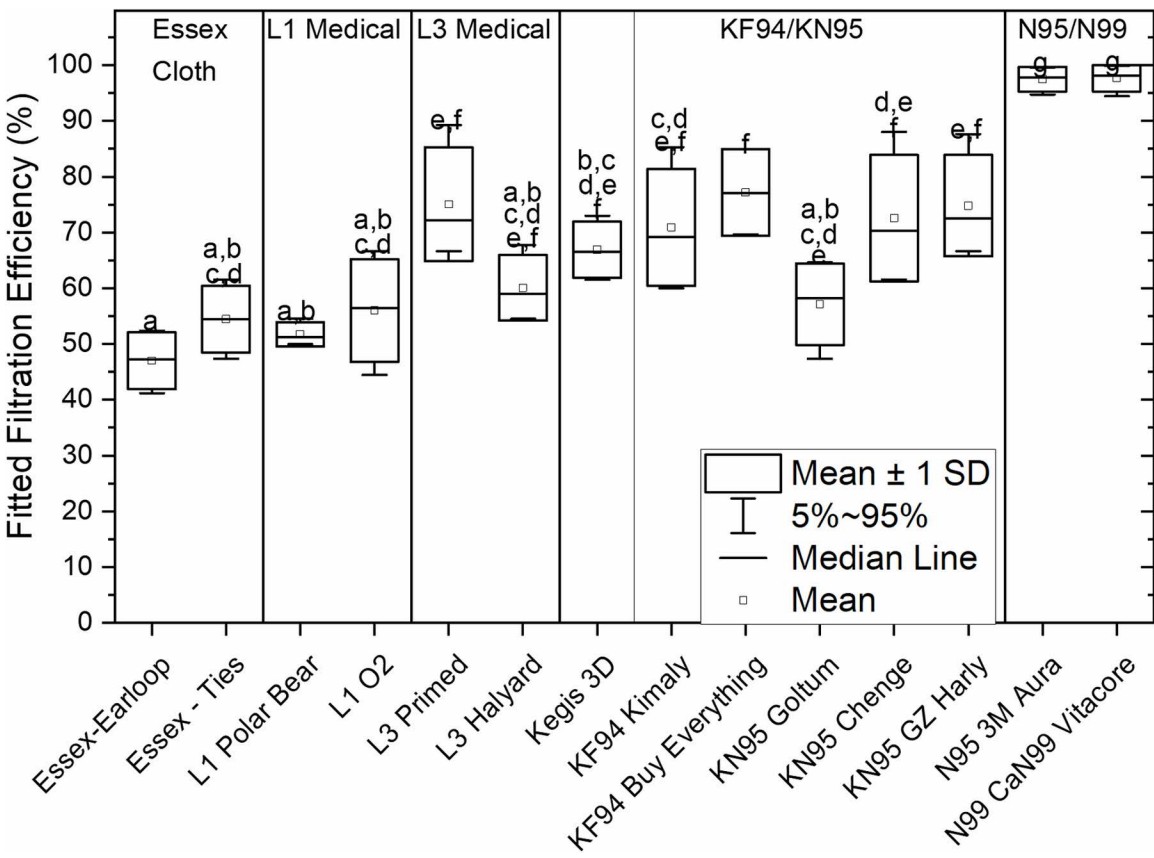

**Fig 1. Fitted filtration efficiency for cloth (2), L1 (2) and L3 (2) certified medical masks, a non-certified Kegis 3D mask (purchased as a KN95 look-alike), KF94s (2), KN95s (3), and for respirators, N95 and CaN99.** N = 4. Bars present mean and standard deviation (SD) and whiskers showing 5 - 95 confidence values. Data were normal by Lillefor's test. Letters above whiskers indicate statistical groupings according to Tukey's post hoc comparisons. A shared letter for two mask types signifies no difference between those types; absence of a shared letter signifies a significant difference p < 0.05.

## Subjective assessment

Data on subjective assessment of leaks, glasses fogging, and discomfort are shown in S2 and S3 Tables and S3–S5 Figs. Leak scores varied by mask type, but not with hacks or overmasking. Glasses fogging showed few statistically significant differences in any substudy. Discomfort scores ranged from 2 (comfortable) to 6 (uncomfortable) and differed across mask/respirator types. By category respirators and medical masks had better comfort than some KF94 and KN95s ($P < 0.05$) but within in each category there were high- and low-comfort masks. Braces, knot-and-tuck and overmasking generated lower comfort compared with controls (all $P < 0.05$). For braces this represented a trade-off between fitted filtration efficiency and comfort. Comfort scores were not related to fitted filtration efficiency.

Subjective assessment of mask leaks and glasses fogging were both significantly related to fitted filtration efficiency ($P < 0.05$, Fig 4). Leak assessment was the stronger predictor, explaining 22% of fitted filtration efficiency, whereas glasses fogging explained 4% of variation. Trials across masks with scores of 1 (no leaks detected) achieved mean ± SD filtration efficiencies of 86 ± 12%, scores of 2 (imperfect seal to face) 74 ± 16%, 3 (minor leaks) 71 ± 15%, 4 (minor leaks in multiple areas) 64 ± 15% and scores between 5–7 (major to severe leaks)

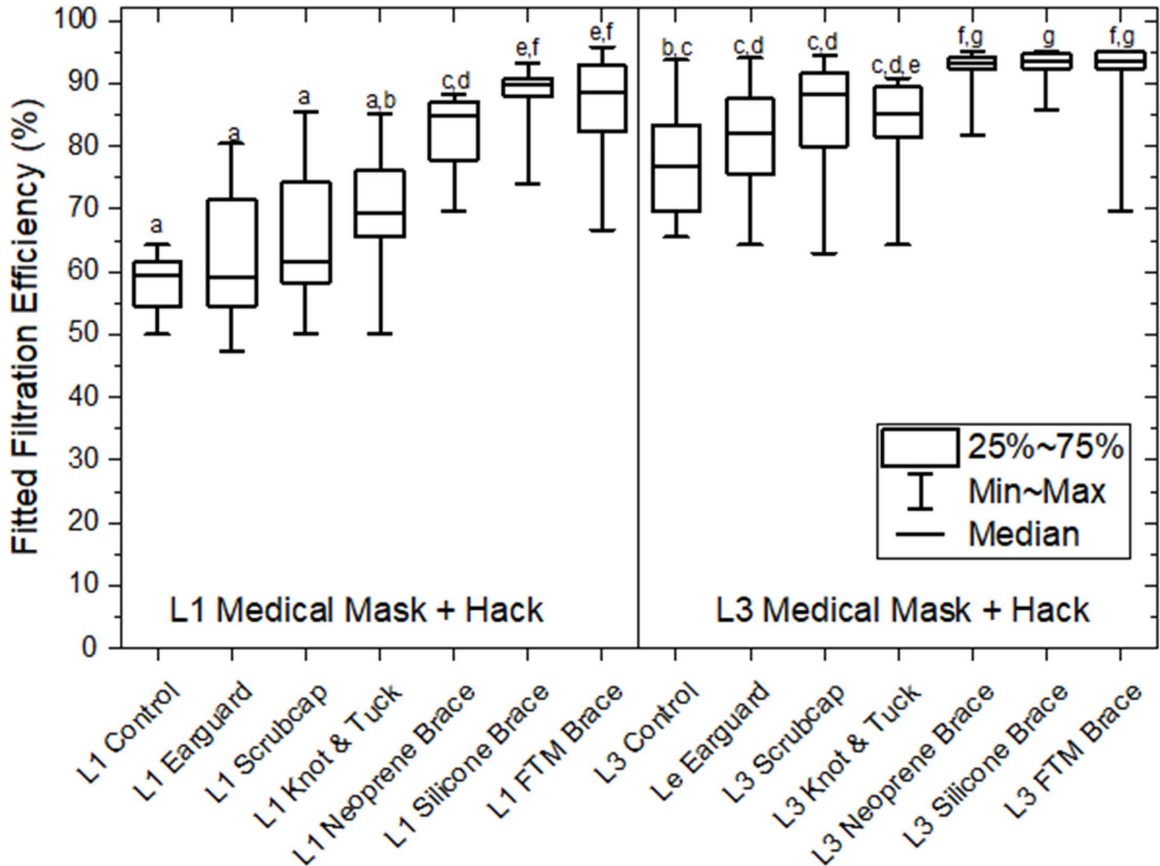

**Fig 2. Box and whisker plot showing the effect of minor modifications, or hacks, to a certified level 1 Polar Bear mask and to a certified level 3 Halyard mask.** 10 participants, 1 replicate. Data were not normal by Lillefor's test. Kruskal-Wallis with Conover-Inman post hoc comparisons. Boxes show interquartile range and whiskers minimum and maximum. Letters denote groups which are statistically similar and dissimilar: a shared letter for two mask types signifies no difference between those types; absence of a shared letter signifies a significant difference p < 0.05. Neoprene brace made using downloadable, public domain, template from Fix The Mask and recommended materials; silicone brace designed at McMaster University; FTM brace - proprietary Fix-The-Mask brace. L1 and L3 controls were retested on these participants as part of this panel; estimates differ slightly from those in Fig 1.

averaged 60 ± 9%. However, these observations were dependent on mask type. For example, among cloth masks tested, the range of FFEs varied by 43% (FFE range of 38–80%) across all trials whereas masks tested and given a score of 1 had mean FFEs that were improved by an average of 7% over the mean FFE of masks with leak scores greater than 5. A similar observation was noted for KF94s and KF95s, with individual masks and trials varying by 35% in FFE performance while leak scores explained about 9% of this performance difference. This indicates difficulties in perceiving leaks for these mask types. Alternatively, for Level 1 and Level 3 masks with and without hacks, perceived leak scores were much stronger predictors of mask performance. Level 1 masks exhibited a range of FFE values that varied by 51% (FFE range from 44 to 96%) across individual trials, whereas trials with Level 1 masks having an assigned leak score of 1 had FFEs between 22 and 26% higher compared to masks of this type given a leak of score of 5 or above. Similarly, leak scores assigned to level 3 masks explained as much as half of the total variation in FFEs scores observed for this mask type. All respirators were assigned the best perceived leak scores of 1 and this mask type showed the least variation in performance among mask types.

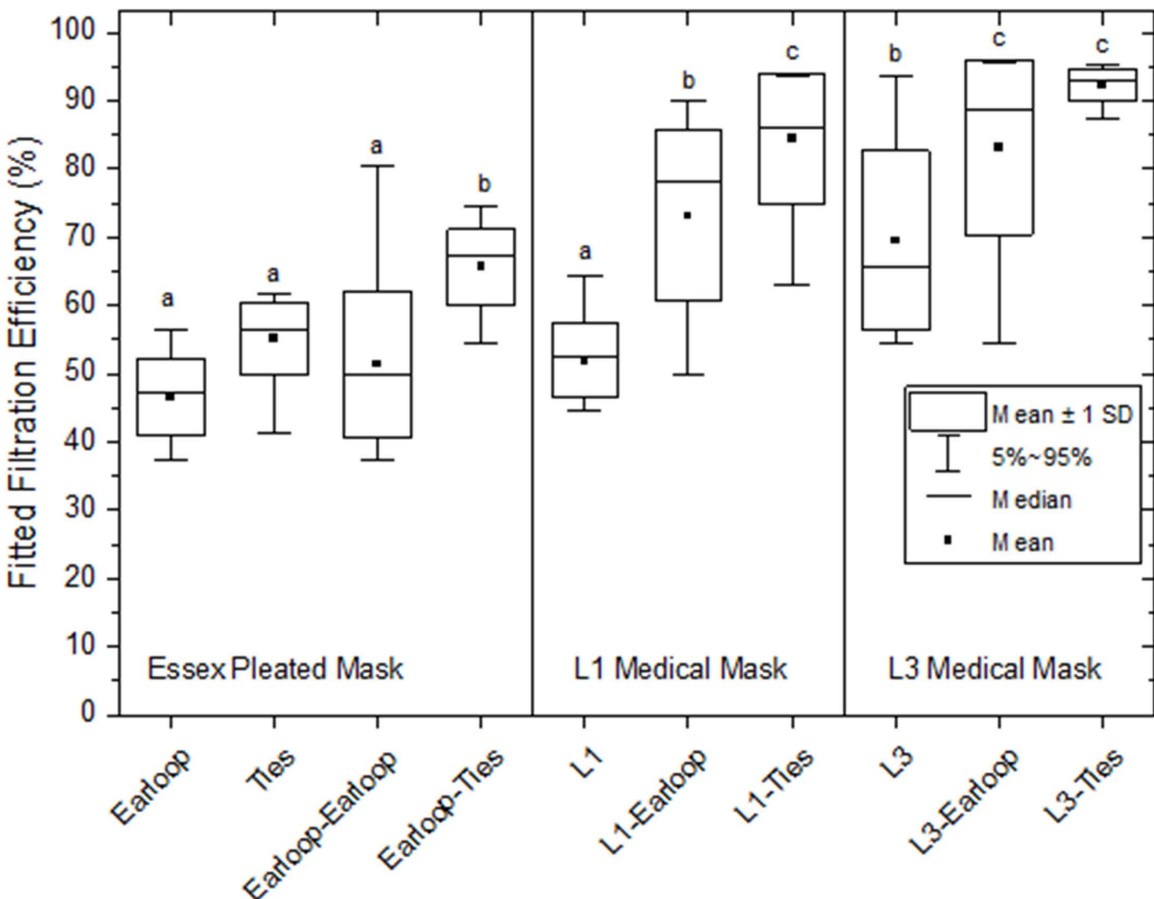

**Fig 3. Effects of overmasking with Essex masks on earloops and on overhead ties on fitted filtration efficiency.** The graph shows (left) the Essex mask on earloops (Earloop) and on ties (Ties) worn alone, followed by Essex-on-earloop with a second Essex-on-earloop as an overmask (Earloop-Earloop), and by Essex-on-earloop with an Essex-on-ties as an overmask (Earloop-Ties). The centre panel shows the level 1 certified Polar Bear mask worn alone (L1), with an Essex-on-earloop as an overmask (L1- Earloop), and with an Essex-on-ties as an overmask (L1-Ties). The right panel shows the level 3 certified Halyard mask worn alone (L3), with an Essex-on-earloop as an overmask (L3-Earloop), and with an Essex-on-ties as an overmask (L3-Ties). 6 participants, 3 replicates. Data were normal by Lillefor's test. ANOVA with Tukey's honestly significant difference for post hoc comparisons was used. Mean and median; boxes show one standard deviation (SD); whiskers show 95% confidence intervals. Letters denote groups which are statistically similar and dissimilar: a shared letter for two mask types signifies no difference between those types; absence of a shared letter signifies a difference. Earloop: Essex mask worn on elastic earloops. Ties: Essex mask worn on overhead cloth ties. L1 level 1; L3 level 3.

## Facial measurements

There was no association between the facial distances bizygomatic distance and menton-sellion length, measured as if for clothing, with a piece of cord that traversed the bridge of the nose and the tip of the nose, respectively, and the same distance measured with calipers: $R^2$ 0.03; $P = 0.53$ and $R^2$ 0.09; $P = 0.27$, respectively (S6 Fig).

## Discussion

We found that well-fitting cotton cloth masks with earloops or ties filtered at 47–55%; this was comparable with level 1 certified masks (51–60%; infographic, S7 Fig). This is not widely known, and not intuitively obvious, because the filtration properties of the materials are very differ-ent [32]. The better edge seal of the well-designed cloth mask makes up for the relatively poor

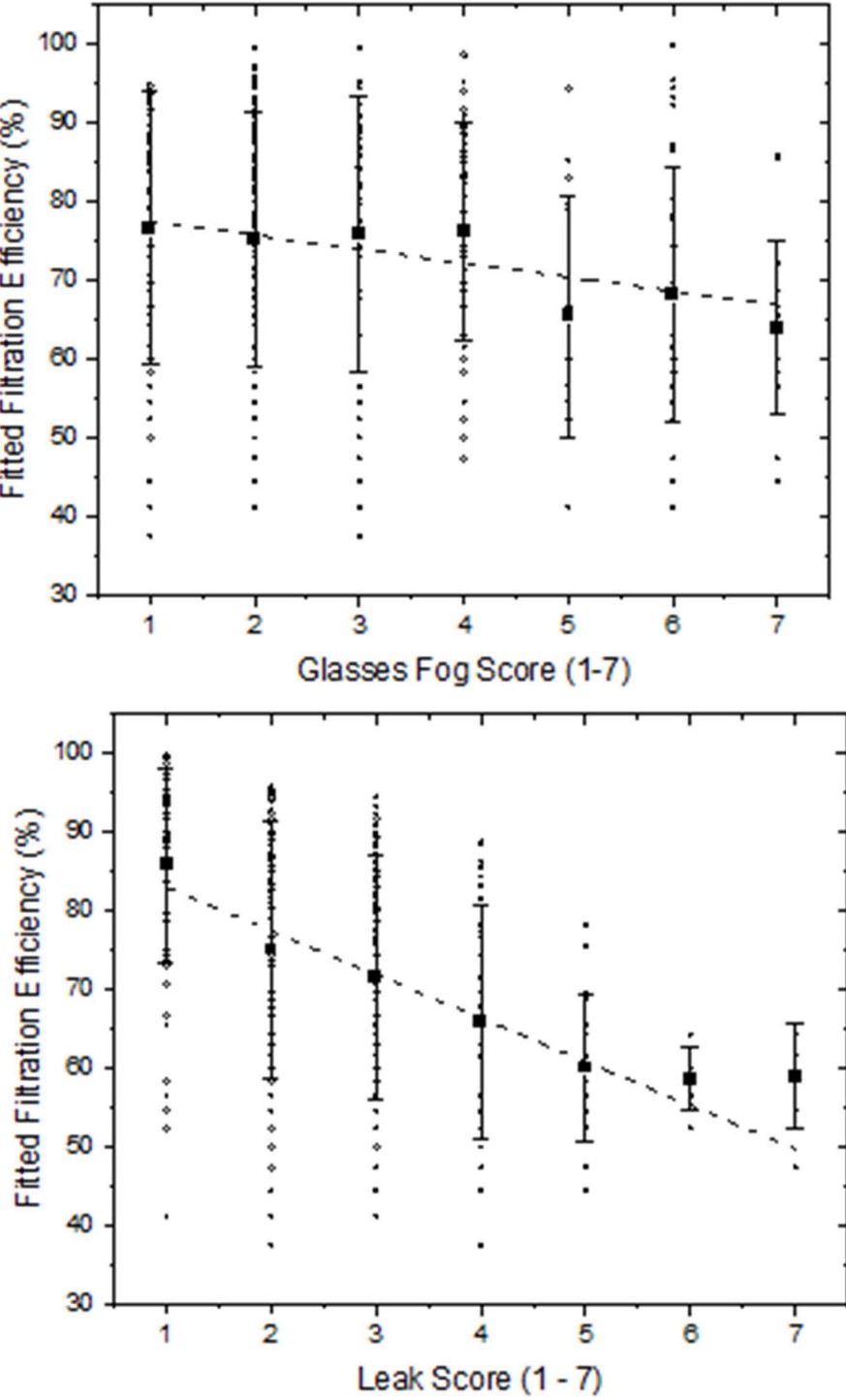

**Fig 4. Relationship between fitted filtration efficiency and glasses fog, and between fitted filtration efficiency and subjective leak.** Glasses fog and subjective leak were assessed before fitted filtration efficiency was measured. Top graphic fitted filtration efficiency against glasses fog score across data generated for each sub-study. Open circles are raw data, squares are means and whiskers are standard deviations for each score category. Dashed line is the linear regression fit: FFE = -1.78 ± 0.48 * Glasses Fog Score + 79.3 ± 1.7; $R^2$ = 0.04; p < 0.001, df = 338. Bottom graphic presents data against leak score. Dashed line regression fit: FFE = -5.5 ± 0.6 * Leak Score + 88.5 ± 1.7; $R^2$ = 0.22; p < 0.001, df = 338.

filtration of woven cotton [7,32]; the excellent filtration (>95%) of the material used in level 1 masks [30,32] is let down by poor fit resulting in edge leak. Level 3 masks were variable (60–77%).

Indeed, participants' blinded a priori numerical assessment of mask leakage accounted for some of the variation in mask performance but this varied between mask types. Assessments about mask leakage were most effective at predicting mask performance for level 1 and level 3 masks, which likely generalizes to all certified medical masks. International standards ensure that material in these masks filters very effectively (> 95%), and reduction in fitted filtration when worn is explained by leak around the mask. The finding that participants' assessment of leak around the mask, for medical masks, explains some of the variability in leak, should be included in public health communication strategies and in instructions about the choice and use of medical masks as PPE. These are among the most commonly used masks by the public, and are widely used in health care in the management of people with COVID-19 in Canada, the UK and the European union [42–44].

In the media [45], in CDC messaging [46], and in epidemiologic research [13], KF94/KN95-type masks are often conflated with N95s, FFP3s, and CaN99s. We observed 57–77% filtration for the KF94/KN95s that we tested. The previous literature is limited to two masks, one with 55% filtration efficiency (7 participants, 1 replicate) [47], and the other 84% (≥8 participants) [48]. We found most, but not all, KN95/KF94s performed better than cloth and level 1 masks, similar to level 3 masks, but not as well as the N95 and CaN99. The material in KF94/KN95s is an excellent filter; the Chengde and the GZHarley masks were tested by NIOSH and passed, with all masks providing greater than 95% filtration when tested with the edges glued to a flat plate in keeping with NIOSH Standard Test Procedure (STP) TEB-APR-STP-0059 [49]. The difference between them when worn must result from design and fit, including the type of head attachment: N95s and CaN99s have overhead elastic.

In contrast, we found 97–98% filtration for N95 and CaN99 respirators that were not formally fit-tested: this is close to the fit-testing threshold of ≥ 99% (i.e., fit factor ≥ 100). Other studies of non-fit-tested respirators have reported 82% to 98% filtration.[47,50–54] Both our respirators were of the novel 3D design, which may fit a wider variety of faces than the cup designs in common use before the pandemic. In terms of the fraction reaching the participant, the respirators let through up to 3% of particles and the KF94/95s up to 43%: a 14-fold difference in exposure. We argue that this difference should be more widely known and that, provided PPE supply for frontline workers is secure, N95s and CaN99 should be recommended for community use.

Many previous studies of fitted filtration efficiency with human participants, for medical and non-medical masks, have been limited by very small sample sizes or incomplete descriptions of the masks. For cloth masks, our data (47–55%) are in keeping with the higher end of reported fitted filtration efficiencies from the following studies: 27% (3-ply cotton on earloops, 0.02–3 μm particles, 1 participant, 4 replicates) [5]; 28% (2-ply, 3-ply and 4-ply polyester, cotton and poly-cotton masks on earloops, 0.1 μm particles; 3–4 participants, 1 replicate) [55]; 50% (2-ply cotton T-shirt fabric, on overhead elastic ties, < 0.1 μm particles, 21 participants, 1 replicate) [56–58]; 52% (head attachments and material not reported, 5 designs including 1-ply and bandana fabric, < 0.1 μm particles, 3 participants, 1 replicate) [47], 60% in adults and 55% in children (TD Cerise Multi teacloth, Blokker; 28 adults and 11 children, 1 replicate) [59]. The filtration we observed, in the 47–55% range, likely reflects the careful design of the Essex mask [33], in keeping with the Davies mask [56,58], and in contrast with random variation from haphazard sampling [5,47,55].

Level 1 masks were associated with filtration of 52–60%, and level 3 masks with filtration of 60–77%. Previous studies have included a single mask, often incompletely specified: 39% (procedure mask on earloops, 0.02–3 μm particles, 1 participant, 4 replicates) [5]; 72% (surgical

mask with ties, 0.02–3 µm particles, 1 participant, 4 replicates) [5]; 69% (unknown design, 0.1 µm particles, 7 participants, 1 replicate) [47]; 80% (level 1 equivalent on ties, 0.1 µm particles, 21 participants) [56], 76% in adults and 69% in children (3M 1818 Tie-On with ties, 0.02–1 µm, 28 adults and 11 children, 1 replicate) [59]. The difference in fitted filtration we observed between some level 3 and level 1 masks has not previously been reported and likely results from better fit: the difference in material flat filtration for these two standards is small (98% v 95% for 0.1 µm latex at the time of this study) [30,31]. The level 3 masks we studied were larger top-to-bottom than the level 1 masks, had foam at the nosepiece, were softer, and appeared to conform better to participants' faces.

Ours is the largest study to date to examine a comprehensive range of minor modifications (mask hacks), and overmasking; and the first to examine the effects on both level 1 and level 3 masks. With a larger sample size than any previous study (10 participants), we were not able to confirm the improvements seen for earguard and knot-and-tuck in previous studies (1 participant, 4 replicates [5]; 3–4 participants [55]), though these methods may increase fitted filtration for some individuals. We confirmed the findings (1 participant, 4 replicates [5]; 11 participants [60] and 3–4 participants [55]) that external braces greatly improve the efficiency of certified masks (infographic, S7 Fig). We have previously shown that braces have little effect over cloth masks [61].

We found that overmasking with a carefully-designed cloth mask was associated with increases in filtration efficiency. Our study of 6 participants with 3 replicates is a significant addition to previous studies (1 participant, 4 replicates, using nylon hosiery [5]; 3–4 participants, overmasking [55]) which also found overmasking effective. Overmasking should be encouraged when N95s are not available.

Our work provides the first direct comparison of earloops and ties, finding ties superior when used as an overmask with a level 1 mask, and a non-significant tendency to superiority when worn alone or over a level 3 mask. This is in keeping with the idea that though earloops are convenient for donning and doffing, overhead attachments produce a closer fit and less edge leak, in addition to the observed differences between KF94/KN95s which have earloops, and N95/CaN99s which have overhead elastic.

Our KN95/KF94 sampling strategy simulated an informed non-expert purchaser: we recognize that our sample is small [62]. It is a limitation of our work that we did not include non-certified pleated disposable masks or CaN95s.

We recognize additional limitations (S6 Appendix). Filtration properties vary with the size and density of particles and it is a limitation that we have no information on this, given that our methodology could not distinguish aerosols present within the test space or interior of the mask contributed by ambient particles from NaCl particles produced by the particle generator. It is possible that the proportion of NaCl and ambient particles varied on different days. However, we note that our methodology is broadly consistent with multiple studies using NaCl particle generators and TSI PortaCount Respirator Fit testers to measure fit factors and mask performance [5,47,56,59,61] We studied one cloth mask, a design that was created and refined by a panel of community experts, optimized, and produced at scale (55,000 masks) by non-expert sewists for distribution to marginalized communities. The ASTM standard for barrier face coverings, F3502, which is a source-control standard, not designed to assess the degree of protection for the wearer, defines filtration efficiency of a mask edge-sealed to a plate, not worn [63]; we did not perform this test. Finally, we examined masks through the lens of wearer protection; we provide no data on source control, for which there is a wealth of data [64] but no standard methodology [63]. However, previous studies suggest a relationship between the two [50,55,65] and F3502 recognizes Portacount testing as the only available quantitative standard for assessing fit for source control [63].

Strengths of our study are the large number of participants compared with previous studies, the characterization of the participants, the characterization of the masks, and the inclusion of a variety of different masks, including KN95/KF94s, and variety of mask hacks, in the same study. We used human participants and studied 0.02 to 1 μm particles, which is at the smaller end of the particle size range that is thought to be most clinically relevant [65–69].

## Conclusions

Well-designed cloth masks exhibited fitted filtration for submicron aerosols similar to that of level 1 masks: around 50%. Level 3 masks and KN95/KF94s performed around 70%. Overmasking of certified level 1 and 3 masks with cloth masks was effective and external braces over level 1 and 3 masks were highly effective in improving filtration efficiency. Filtration at or close to occupational health standards for PPE was observed only for N95s and CaN99s.

Filtration is size and weight dependent. Current standards for protection from bio-aerosols derive from those for industrial protection: N95 respirators are certified and used in both contexts. The standards for N95s are 95% filtration of dense salt particles of 0.075 μm count median diameter for the mask sealed to a plate, followed by fit test-ing on humans with the N95 classifier engaged, reflecting particles as small as 0.2 μm in diameter [70–72]. These standards are generally met by constructing masks from melt-blown electrostatically-modified polypropylene or other petroleum-derived plastics. Viruses, mucus and water are less dense than salt: bioaerosols are often assumed to have density similar to water at $1\,g/cm^3$ [73], whereas the density of salt is $2.2\,g/cm^3$ [74]. (Salt particles are generated as saline but dehydrate to solid salt within milliseconds [72,75].) Though the rigorous standards adopted from industry translate into safety for health care workers, when applied, it is not clear that these are the optimal health care standards, and new standards, equally safe, could be developed based on the filtration properties of clinically-important bioaerosols. Such re-engineering would also facilitate improvements in breathability and perhaps in sustainability, and further research in this are would also be applicable to masks worn to reduce the transmission of respiratory diseases in the commu-nity (i.e., barrier face coverings).

Cloth masks and medical masks, widely used in the pandemic, filter aerosols, account-ing for their partial effectiveness in reducing transmission in an airborne pandemic [9,13,76–79]; this should inform public health messaging and advice. Medical masks are no substitute for respirators in the personal protection of health care workers, in keeping with meta-analysis of trials comparing continuous mask use with continuous respirator use [17].

## Supporting information

**S1 Table. Characteristics of the 12 participants, including ethnic origin, facial hair type and NIOSH panel size.** * Face width narrower than smallest face width on the NIOSH panel, classified according to menton-sellion. Facial hair (mask hacks study only) was classified according to CDC NIOSH (24).
(PDF)

**S2 Table. Subjective data for the mask hacks substudy; top panel, level 1 ASTM certified mask; bottom panel, level 3 ASTM certified mask. Leaks, Glasses Fog and Comfort are also shown graphically in S3-5 Figs.** *1 Moving into eyes, 2 Moving around, not into eyes, 3 Straps pinching, 4 Nose stuffy, 5 Hard to breathe, 6 Interferes with hair/head covering;

multiple occurrences of the same number indicate that multiple participants reported the same issue. FTM Fix the Mask.
(PDF)

**S3 Table. Subjective data for the overmasking study; top panel, level 1 and level 3 ASTM certified mask. Leaks, Glasses Fog and Comfort are shown graphically in S3-5 Figs.** *1 Moving into eyes, 2 Moving around, not into eyes, 3 Straps pinching, 4 Nose stuffy, 5 Hard to breathe, 6 Interferes with hair/head covering Multiple occurrences of the same number indicate that multiple participants reported the same issue. L1 level 1; L3 level 3; EL Earloops; FT Fabric ties.
(PDF)

**S1 Fig. Descriptions and photographs of representative masks and of minor modifications (hacks) to improve fitted filtration efficiency of masks.**
(PDF)

**S2 Fig. A research participant wearing a cloth mask on ties for testing.**
(PDF)

**S3 Fig. Subjective leak scores. Normal when mean shown, otherwise non-normal. Top panel mask types; middle panel mask hacks; bottom panel overmasking.**
(PDF)

**S4 Fig. Subjective glasses fog scores. Normal when mean shown, otherwise non-normal. Top panel mask types; middle panel mask hacks; bottom panel overmasking.**
(PDF)

**S5 Fig. Subjective discomfort scores. Normal when mean shown, otherwise non-normal. Top panel mask types; middle panel mask hacks; bottom panel overmasking.**
(PDF)

**S6 Fig. Relationship between cord measures and caliper measures, for bizygomatic and for menton-sellion distances, R2 and p for linear regression; line of best fit not shown because P > 0.05 and R2 small.**
(PDF)

**S7 Fig. Infographic summarizing study results.**
(JPG)

**S1 Appendix. Facial Measurement: General.**
(PDF)

**S2 Appendix. KN95, KF94 and non-certified mask purchase.**
(PDF)

**S3 Appendix. Rationale: filtration efficiency testing.**
(PDF)

**S4 Appendix. Qualitative Data Collection.**
(PDF)

**S5 Appendix. Cloth Mask Knowledge Exchange.**
(PDF)

**S6 Appendix. Limitations and areas for future research.**
(PDF)

## Key Points

Question: How well do medical and non-medical masks filter aerosols when worn?

Findings: Well-fitting 2-layer cotton masks, and level 1 medical masks were similar, both filtering around 50% of aerosols. Level 3 masks and KN95/KF94s were similar, filtering around 70%. N95s and CaN99s, without formal fit testing, filtered 97–98%.

Meaning: Level 1 medical masks were not better than the well-fitting 2-layer cotton masks we tested. KN95/KF94s are not as efficient, when worn, as N95s and CaN99s. Overmasking and the use of external braces improve filtration: these are potentially useful strategies when N95s are not available.

See infographic, S7 Fig. Available for free download, along with low-ink and black-and-white versions at https://maskevidence.org/masks-compared. Thumbnail: (S7 Fig)

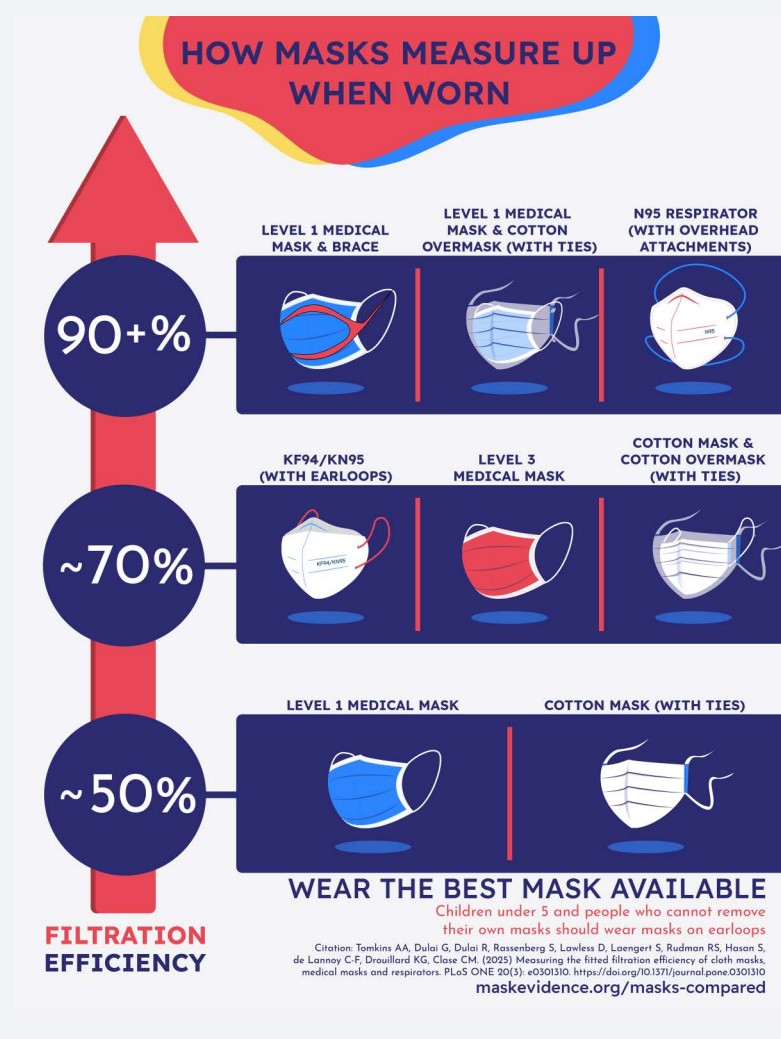

## Acknowledgements

The Centre of Excellence in Protective Equipment and Materials and McMaster University are built on the traditional territory of the Haudenosaunee and Anishinaabe first nations, recognized in the 1701 Dish with One Spoon Wampum.

We thank Bryan Herechuk, Jeff Mallany, Anna Dorey, Seema Sharma, Hamilton Health Sciences, Hamilton, St Joseph's Healthcare Hamilton, Hamilton, ON for mask donations, materials, advice and encouragement.

Windsor Essex Sewing Force (WESF) provided cloth masks, advice and encouragement. They place their pattern in the public domain with this publication [33].

We thank the stakeholder group Cloth Mask Knowledge Exchange (CMKE) for advice and encouragement at every stage of this work. We are particularly grateful to Patricia Savage for her contribution in sewing standardized cloth masks.

We dedicate this work to the memory of our CMKE colleague Helen Brunet.

## Author contributions

**Conceptualization:** Amanda A Tomkins, Gurleen Dulai, Ranmeet Dulai, Sarah Rassenberg, Darren Lawless, Scott Laengert, Rebecca S Rudman, Charles-Francois de Lannoy, Ken G Drouillard, Catherine M Clase.

**Data curation:** Amanda A Tomkins, Gurleen Dulai, Ken G Drouillard, Catherine M Clase.

**Formal analysis:** Amanda A Tomkins, Gurleen Dulai, Ranmeet Dulai, Ken G Drouillard, Catherine M Clase.

**Investigation:** Amanda A Tomkins, Gurleen Dulai, Ranmeet Dulai, Sarah Rassenberg, Scott Laengert, Rebecca S Rudman, Charles-Francois de Lannoy, Ken G Drouillard, Catherine M Clase.

**Methodology:** Amanda A Tomkins, Gurleen Dulai, Ranmeet Dulai, Sarah Rassenberg, Scott Laengert, Rebecca S Rudman, Charles-Francois de Lannoy, Ken G Drouillard, Catherine M Clase.

**Project administration:** Darren Lawless, Rebecca S Rudman, Charles-Francois de Lannoy, Ken G Drouillard, Catherine M Clase.

**Resources:** Gurleen Dulai, Ranmeet Dulai, Darren Lawless, Rebecca S Rudman, Charles-Francois de Lannoy, Ken G Drouillard, Catherine M Clase.

**Software:** Ken G Drouillard, Catherine M Clase.

**Supervision:** Charles-Francois de Lannoy, Ken G Drouillard, Catherine M Clase.

**Validation:** Amanda A Tomkins, Gurleen Dulai, Ranmeet Dulai, Ken G Drouillard, Catherine M Clase.

**Visualization:** Amanda A Tomkins, Ranmeet Dulai, Rebecca S Rudman, Shiblul Hasan, Ken G Drouillard, Catherine M Clase.

**Writing – original draft:** Amanda A Tomkins, Ken G Drouillard, Catherine M Clase.

**Writing – review & editing:** Amanda A Tomkins, Gurleen Dulai, Ranmeet Dulai, Sarah Rassenberg, Darren Lawless, Scott Laengert, Rebecca S Rudman, Shiblul Hasan, Charles-Francois de Lannoy, Ken G Drouillard, Catherine M Clase.

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
