## [Decision Letter · Decision Letter 0]

31 May 2024

PONE-D-24-09859Measuring the fitted filtration efficiency of cloth masks, medical masks and respirators

PLOS ONE

Dear Dr. Clase,

Thank you for submitting your manuscript to PLOS ONE. After careful consideration, we feel that it has merit but does not fully meet PLOS ONE’s publication criteria as it currently stands. Therefore, we invite you to submit a revised version of the manuscript that addresses the points raised during the review process.

The work is interested, however, reviewers have given comments for improvement. E.g.  It should be mentioned that FE is based on total measured particle count and size-dependent.

We look forward to receiving your revised manuscript.

Kind regards,

Yasir Nawab, PhD

Academic Editor

PLOS ONE

2. Thank you for stating the following in the Competing Interests section: "Amanda Tomkins is a member of Dr Qiyin Fang’s research group which worked on the silicone mask brace. She is also a member of the cloth mask knowledge exchange, a stakeholder group that includes cloth mask manufacturers and fabric distributors.

Catherine Clase has received consultation, advisory-board membership, honoraria, or research funding  from  the  Ontario  Ministry  of  Health, Sanofi, Pfizer,  Leo  Pharma, Astellas,  Janssen,  Amgen,  Boehringer-Ingelheim,  Baxter and, through LiV Academy, AstraZeneca. In 2018 she co-chaired a KDIGO potassium controversies conference sponsored at arm's length by Fresenius Medical Care, AstraZeneca, Vifor Fresenius Medical Care, Relypsa, Bayer HealthCare and Boehringer Ingelheim. She co-chairs the cloth mask knowledge exchange, a stakeholder group that includes cloth mask manufacturers and fabric distributors. She is editor-in-chief of MaskEvidence.org.

Ken G Drouillard is a member of the WE-SPARK Health Institute, University of Windsor and receives funding from the Natural Sciences and Engineering Research Council of Canada, Environment and Climate Change Canada and Ontario Ministry of Conservation, Environment and Parks. In 2020-2022 he acted as science consultant to the Windsor-Essex Sewing Force, a community group engaged in the design, sewing and donation of cloth masks to healthcare providers and vulnerable populations of Southern Ontario. He is a member of the cloth mask knowledge exchange.

Charles-Francois de Lannoy has received funding from various branches of The Natural Sciences and Engineering Research Council of Canada (NSERC), Ontario Centre of Innovation (OCI), formerly Ontario Centres of Excellence (OCE), Ontario Water Consortium (OWC) formerly Southern Ontario Water Consortium (SOWC), Canada First Research Excellence Fund (CFREF), Ontario Together Fund, and Federal Economic Development Agency for Southern Ontario (FedDev). He is a member of cloth mask knowledge exchange, a stakeholder group that includes cloth mask manufacturers and fabric distributors. 

Darren Lawless co-chairs the cloth mask knowledge exchange, and all authors are members.

Other authors have no additional disclosures.

" 

Reviewers' comments:

Reviewer's Responses to Questions

**Comments to the Author**

1. Is the manuscript technically sound, and do the data support the conclusions?

Reviewer #1: Yes

Reviewer #2: Yes

2. Has the statistical analysis been performed appropriately and rigorously? 

Reviewer #1: Yes

Reviewer #2: Yes

3. Have the authors made all data underlying the findings in their manuscript fully available?

Reviewer #1: Yes

Reviewer #2: Yes

4. Is the manuscript presented in an intelligible fashion and written in standard English?

Reviewer #1: Yes

Reviewer #2: Yes

5. Review Comments to the Author

Reviewer #1: This manuscript excellently explains the experimental conduct by the authors.

I would recommend to accept it in current condition. Just minor revision to english proficiency.

The manuscript is good to accept.

Reviewer #2: Comments to the Author

Measuring the fitted filtration efficiency (FFE) of cloth masks, medical masks, and respirators for small aerosol (0.02 – 1 µm) and effect on FFE by user modifications, over masking etc. The study is quite interesting and explained well. However, the manuscript needs to be revised before consideration. The section-wise comments/suggestions on the article are given below:

Abstract:

Closed room with ambient particles supplemented with salt particles. It would be better if a total fraction of ambient and salt aerosol were given. The FFE need to be written with a standard deviation like FFE ± SD.

Introduction:

The introduction is written shortly and concisely; however, some recent literature was missing in a similar domain, which can also be included in the main manuscript or supplementary file. A few are given below:

• Quantitative performance analysis of respiratory facemasks using atmospheric and laboratory generated aerosols following with gamma sterilization. Aerosol and Air Quality Research, 21(1), 200349.

• Evaluation of filtration effectiveness of various types of facemasks following with different sterilization methods. Journal of Industrial Textiles, 51(2_suppl), 3430S-3465S.

• A detailed investigation of N95 respirator sterilization with dry heat, hydrogen peroxide, and ionizing radiation. Journal of Industrial Textiles, 51(1_suppl), 378S-405S.

Methods:

Page 5: How long does a generated NaCl aerosol drying in a room environment? The room temperature and humidity also play a role. The temperature and humidity may be given in the manuscript.

The particle counts between 2000 to 20,000 in per cm3 or per litter or per m3.

Page 6: The statistical methods section needs to be revised into simple sentences. Entire paragraph is given in one sentence. It is tough to understand.

Results:

Table 1: mean and SD are given as comma-separated; it should be better if it is given FE ± SD.

Supplementary file Fig. 1: A two-layer pleated mask should have pleated counts in each mask type. Levels 1, and 3 and certified masks should have detailed specifications like material density, fibre diameter, nose clip, breathing resistance etc.

Supplementary file page19: Given the well-accepted U-shaped relationship between particle size and filtration with a nadir around 0.3 µm,23 it seems probable that our choice of particle range (0.02 – 1 µm) leads to estimates that are conservative compared with those obtained by testing only smaller particles or including much larger particles. Here the sentence should be modified because many studies have been conducted for particle ranges from 10 nm to 10 µm.

Supplementary file page 22: In the Glass fog rating scale, rating numbers 3 & 5 are blank. Something missing?

Supplementary file page 36: Limitation and area for future research: many works have been already performed for many tabulated points. However, agreed with the authors, many areas need to be further researched.

Discussion:

Page 10: Previous studies of FFE for medical and non-medical masks have been limited by minimal sample size and incomplete description of the mask. Please see each detail given for FE and mask characterization in the study below. Hence the sentence may be modified accordingly.

• Quantitative performance analysis of respiratory facemasks using atmospheric and laboratory generated aerosols following with gamma sterilization. Aerosol and Air Quality Research, 21(1), 200349.

• Evaluation of filtration effectiveness of various types of facemasks following with different sterilization methods. Journal of Industrial Textiles, 51(2_suppl), 3430S-3465S.

• A detailed investigation of N95 respirator sterilization with dry heat, hydrogen peroxide, and ionizing radiation. Journal of Industrial Textiles, 51(1_suppl), 378S-405S.

Page 11: The filtration efficiency should also have a standard deviation in running text, though is given in figures.

Page 11: some comments can be added on the filtration efficiency of various masks without leakage.

Conclusion:

In conclusion,, it should be mentioned that FE is based on total measured particle count and size-dependent.

6. PLOS authors have the option to publish the peer review history of their article (what does this mean? ). If published, this will include your full peer review and any attached files.

**Do you want your identity to be public for this peer review?** For information about this choice, including consent withdrawal, please see our Privacy Policy .

Reviewer #1: **Yes: ** Syed Muhammad Zaigham Abbas Naqvi

Reviewer #2: **Yes: ** Dr. Amit Kumar

---

## [Author Response · Author response to Decision Letter 1]

15 Jul 2024

Response to reviews 2024-06-26

PONE-D-24-09859

Measuring the fitted filtration efficiency of cloth masks, medical masks and respirators

PLOS ONE

Dear Dr. Clase,

Thank you for submitting your manuscript to PLOS ONE. After careful consideration, we feel that it has merit but does not fully meet PLOS ONE’s publication criteria as it currently stands. Therefore, we invite you to submit a revised version of the manuscript that addresses the points raised during the review process.

The work is interested, however, reviewers have given comments for improvement. E.g. It should be mentioned that FE is based on total measured particle count and size-dependent.

We look forward to receiving your revised manuscript.

Kind regards,

Yasir Nawab, PhD

Academic Editor

PLOS ONE

Done, many thanks for providing those templates.

2. Thank you for stating the following in the Competing Interests section: "Amanda Tomkins is a member of Dr Qiyin Fang’s research group which worked on the silicone mask brace. She is also a member of the cloth mask knowledge exchange, a stakeholder group that includes cloth mask manufacturers and fabric distributors.

Catherine Clase has received consultation, advisory-board membership, honoraria, or research funding from the Ontario Ministry of Health, Sanofi, Pfizer, Leo Pharma, Astellas, Janssen, Amgen, Boehringer-Ingelheim, Baxter and, through LiV Academy, AstraZeneca. In 2018 she co-chaired a KDIGO potassium controversies conference sponsored at arm's length by Fresenius Medical Care, AstraZeneca, Vifor Fresenius Medical Care, Relypsa, Bayer HealthCare and Boehringer Ingelheim. She co-chairs the cloth mask knowledge exchange, a stakeholder group that includes cloth mask manufacturers and fabric distributors. She is editor-in-chief of MaskEvidence.org.

Ken G Drouillard is a member of the WE-SPARK Health Institute, University of Windsor and receives funding from the Natural Sciences and Engineering Research Council of Canada, Environment and Climate Change Canada and Ontario Ministry of Conservation, Environment and Parks. In 2020-2022 he acted as science consultant to the Windsor-Essex Sewing Force, a community group engaged in the design, sewing and donation of cloth masks to healthcare providers and vulnerable populations of Southern Ontario. He is a member of the cloth mask knowledge exchange.

Charles-Francois de Lannoy has received funding from various branches of The Natural Sciences and Engineering Research Council of Canada (NSERC), Ontario Centre of Innovation (OCI), formerly Ontario Centres of Excellence (OCE), Ontario Water Consortium (OWC) formerly Southern Ontario Water Consortium (SOWC), Canada First Research Excellence Fund (CFREF), Ontario Together Fund, and Federal Economic Development Agency for Southern Ontario (FedDev). He is a member of cloth mask knowledge exchange, a stakeholder group that includes cloth mask manufacturers and fabric distributors.

Darren Lawless co-chairs the cloth mask knowledge exchange, and all authors are members.

Other authors have no additional disclosures.

"

Reviewed, unchanged.

We added a sentence on data sharing to Results, paragraph 1:

The 12 participants were 21 – 55 years, 58% female, 25% non-European, NIOSH 1 to 10 (table and supplementary table 1). Data are shared in an online repository at https://osf.io/58g2j/.

We added the declaration above at the end of the Acknowledgements, please let us know if this wasn’t correct.

This does not alter our adherence to PLOS ONE policies on sharing data and materials.

No updates needed

Thank you for this. We have uploaded the data to https://osf.io/58g2j/

Done, thanks for the detailed instructions

Reviewers' comments:

Reviewer's Responses to Questions

Comments to the Author

1. Is the manuscript technically sound, and do the data support the conclusions?

Reviewer #1: Yes

Reviewer #2: Yes

2. Has the statistical analysis been performed appropriately and rigorously?

Reviewer #1: Yes

Reviewer #2: Yes

3. Have the authors made all data underlying the findings in their manuscript fully available?

Reviewer #1: Yes

Reviewer #2: Yes

4. Is the manuscript presented in an intelligible fashion and written in standard English?

Reviewer #1: Yes

Reviewer #2: Yes

5. Review Comments to the Author

Reviewer #1: This manuscript excellently explains the experimental conduct by the authors.

I would recommend to accept it in current condition. Just minor revision to english proficiency.

Thanks to this reviewer for their very generous comments and positive recommendations.

The manuscript is good to accept.

Reviewer #2: Comments to the Author

Measuring the fitted filtration efficiency (FFE) of cloth masks, medical masks, and respirators for small aerosol (0.02 – 1 µm) and effect on FFE by user modifications, over masking etc. The study is quite interesting and explained well. However, the manuscript needs to be revised before consideration. The section-wise comments/suggestions on the article are given below:

Abstract:

Closed room with ambient particles supplemented with salt particles. It would be better if a total fraction of ambient and salt aerosol were given.

Absolutely, we agree. Another better approach is to conduct the experiments in a sealed chamber, so that only salt particles are in play. The equipment to measure the particle types and sizes, and a sealed chamber were not available at our centre. We think that ambient particles are likely a good surrogate for bioaerosol, and salt particles likely lead to underestimation of filtration, compared with less-dense bioaerosols, so that the results we’ve obtained for filtration are conservative.

We’ve added this issue as a limitation to the Discussion:

We recognize additional limitations (S6 Appendix). Filtration properties vary with the size and density of particles and it is a limitation that we have no information on this given that our methodology could not delineate aerosols present within the test space or interior of the mask contributed by ambient particles from NaCl particles produced by the particle generator. It is possible that the proportion of NaCl and ambient particles varied on different days. However, we note that our methodology is broadly consistent with multiple studies using NaCl particle generators and TSI PortaCount Respirator Fit testers to measure fit factors and mask performance (5, 45, 54, 57, 59).

The FFE need to be written with a standard deviation like FFE ± SD.

Done, throughout

Introduction:

The introduction is written shortly and concisely; however, some recent literature was missing in a similar domain, which can also be included in the main manuscript or supplementary file. A few are given below:

• Quantitative performance analysis of respiratory facemasks using atmospheric and laboratory generated aerosols following with gamma sterilization. Aerosol and Air Quality Research, 21(1), 200349.

• Evaluation of filtration effectiveness of various types of facemasks following with different sterilization methods. Journal of Industrial Textiles, 51(2_suppl), 3430S-3465S.

• A detailed investigation of N95 respirator sterilization with dry heat, hydrogen peroxide, and ionizing radiation. Journal of Industrial Textiles, 51(1_suppl), 378S-405S.

We added the issue of sterilization, and these references, to S6 Appendix:

7. No information on washability of masks or sterilization of respirators (2-4)

Methods:

Page 5: How long does a generated NaCl aerosol drying in a room environment? The room temperature and humidity also play a role. The temperature and humidity may be given in the manuscript.

Aerosols of this size dry in milliseconds in a room environment. We added this sentence to the discussion:

(Salt particles are generated as saline but dehydrate to solid salt within milliseconds (71, 74).

The particle counts between 2000 to 20,000 in per cm3 or per litter or per m3.

Our apologies, we have added the units. This sentence now reads:

We proceeded with testing if the total particle count was between 2000 and 20,000 per cm2.

Page 6: The statistical methods section needs to be revised into simple sentences. Entire paragraph is given in one sentence. It is tough to understand.

Sorry about that. We have broken this up, it now reads:

We used SYSTAT v13, Inpixon, Palo Alto, CA for statistical analyses. We analyzed the transformed variable filtration efficiency. We used parametric tests (ANOVA with post hoc Tukey’s honestly significant difference) when data met normality assumptions by Lilliefors test. We used non-parametric tests (Kruskal-Wallis with post-hoc Conover-Inman pairwise comparisons) when they did not, estimating mean and 95% confidence intervals or median and interquartile range, respectively. Throughout, we regarded p<0.05 as statistically significant.

Results:

Table 1: mean and SD are given as comma-separated; it should be better if it is given FE ± SD.

Done.

Supplementary file Fig. 1: A two-layer pleated mask should have pleated counts in each mask type. Levels 1, and 3 and certified masks should have detailed specifications like material density, fibre diameter, nose clip, breathing resistance etc.

We have added the pleat numbers and a webreference for the detailed pattern for the cloth mask. We have added information about the nosewires and the physical structure of the commercial masks. We are sorry, we do not have information on breathing resistance or fibre diameters.

Supplementary file page19: “Given the well-accepted U-shaped relationship between particle size and filtration with a nadir around 0.3 µm,23 it seems probable that our choice of particle range (0.02 – 1 µm) leads to estimates that are c

---

## [Decision Letter · Decision Letter 1]

18 Sep 2024

PONE-D-24-09859R1Measuring the fitted filtration efficiency of cloth masks, medical masks and respiratorsPLOS ONE

Dear Dr. Clase,

Thank you for submitting your manuscript to PLOS ONE. After careful consideration, we feel that it has merit but does not fully meet PLOS ONE’s publication criteria as it currently stands. Therefore, we invite you to submit a revised version of the manuscript that addresses the points raised during the review process.

**Please respond to the comments of reviewer 3.**

We look forward to receiving your revised manuscript.

Kind regards,

Simanta Roy

Academic Editor

PLOS ONE

**Journal Requirements:**

Reviewers' comments:

Reviewer's Responses to Questions

**Comments to the Author**

1. If the authors have adequately addressed your comments raised in a previous round of review and you feel that this manuscript is now acceptable for publication, you may indicate that here to bypass the “Comments to the Author” section, enter your conflict of interest statement in the “Confidential to Editor” section, and submit your "Accept" recommendation.

Reviewer #1: All comments have been addressed

Reviewer #3: All comments have been addressed

2. Is the manuscript technically sound, and do the data support the conclusions?

Reviewer #1: Yes

Reviewer #3: Yes

3. Has the statistical analysis been performed appropriately and rigorously? 

Reviewer #1: I Don't Know

Reviewer #3: Yes

4. Have the authors made all data underlying the findings in their manuscript fully available?

Reviewer #1: Yes

Reviewer #3: Yes

5. Is the manuscript presented in an intelligible fashion and written in standard English?

Reviewer #1: Yes

Reviewer #3: Yes

6. Review Comments to the Author

**Reviewer #1:**  The manuscript found to be correct and all the comments have been discussed by authors. Manuscript can ve published in current format.

**Reviewer #3:**  Measuring the fitted filtration efficiency (FFE) of cloth masks, medical masks, and respirators for small aerosol (0.02 – 1 μm) and effect on FFE by user modifications, over masking etc. The study is relevant and explained well. However, the manuscript needs to be minor revised before consideration. The section-wise

comments/suggestions on the article are given below:

-Introduction are written well. Concise. Added adequate references. However, you can review these articles also. https://doi.org/10.1177/00405175211046056,
https://doi.org/10.3390/ijerph19116372

-On Methods section- Procedures, the authors discussed Filtration efficiency testing. However, what the authors did with the cloth masks are not clear and it is hard to understand by checking with the figure S2. It might be easy to follow if they can denote the things in the figure. Which things refers to what?

Rest of the sections, they wrote well and well described.

However, this paper has to go through some minor revisions, such as

- they wrote “There was no association between the facial distances bizygomatic distance and menton-sellion length, measured as if for clothing, with a piece of cord that traversed the bridge of the nose and the tip of the nose, respectively, and the same distance measured with calipers: R2 0.03; p=0.53 and R2 0.09; p=0.27, respectively (S6 Fig).” Here, this is not “p”, this should be written in italic form.

7. PLOS authors have the option to publish the peer review history of their article (what does this mean? ). If published, this will include your full peer review and any attached files.

**Do you want your identity to be public for this peer review?** For information about this choice, including consent withdrawal, please see our Privacy Policy .

Reviewer #1: **Yes: ** Syed Muhammad Zaigham Abbas Naqvi

Reviewer #3: No

---

## [Author Response · Author response to Decision Letter 2]

3 Feb 2025

Response to reviewer 3

Reviewer #3: Measuring the fitted filtration efficiency (FFE) of cloth masks, medical masks, and respirators for small aerosol (0.02 – 1 μm) and effect on FFE by user modifications, over masking etc. The study is relevant and explained well. However, the manuscript needs to be minor revised before consideration.

Many thanks for your kind comments.

The section-wise

comments/suggestions on the article are given below:

-Introduction are written well. Concise. Added adequate references. However, you can review these articles also. https://doi.org/10.1177/00405175211046056,
https://doi.org/10.3390/ijerph19116372

We added both of these in the introduction, thank you for bringing them to our attention. We also added a recent multidisciplinary review article.

-On Methods section- Procedures, the authors discussed Filtration efficiency testing. However, what the authors did with the cloth masks are not clear and it is hard to understand by checking with the figure S2. It might be easy to follow if they can denote the things in the figure. Which things refers to what?

Thank you for this suggestion. We have labelled figure S2 appropriately, showing the cloth mask on ties, the probe, the tubing and the Portacount device.

Rest of the sections, they wrote well and well described.

We appreciate this kind comment.

However, this paper has to go through some minor revisions, such as

- they wrote “There was no association between the facial distances bizygomatic distance and menton-sellion length, measured as if for clothing, with a piece of cord that traversed the bridge of the nose and the tip of the nose, respectively, and the same distance measured with calipers: R2 0.03; p=0.53 and R2 0.09; p=0.27, respectively (S6 Fig).” Here, this is not “p”, this should be written in italic form.

Thank you. We have replaced small p with capital P, italicised, throughout:

Many thanks for your review and your positive appraisal of our work; and thanks to the editorial team for all the work to date. Please let us know if you have further questions about our work.

---

## [Editor Report · Decision Letter 2]

16 Feb 2025

Measuring the fitted filtration efficiency of cloth masks, medical masks and respirators

PONE-D-24-09859R2

Dear Dr. Clase,

We’re pleased to inform you that your manuscript has been judged scientifically suitable for publication and will be formally accepted for publication once it meets all outstanding technical requirements.

Kind regards,

Simanta Roy

Academic Editor

PLOS ONE
---

## [Editor Report · Acceptance letter]

PONE-D-24-09859R2

PLOS ONE

Dear Dr. Clase,

I'm pleased to inform you that your manuscript has been deemed suitable for publication in PLOS ONE. Congratulations! Your manuscript is now being handed over to our production team.

Kind regards,

on behalf of

Dr. Simanta Roy

Academic Editor

PLOS ONE